# OnlyFlow: Optical Flow based Motion Conditioning for Video Diffusion Models

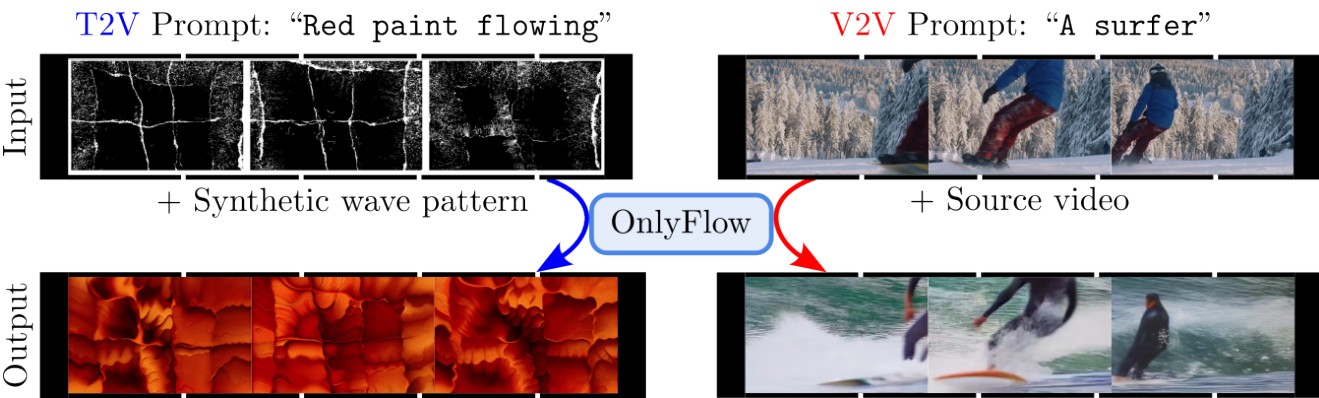

T2V Prompt: "Red paint flowing"

V2V Prompt: "A surfer"

Input

+ Synthetic wave pattern

OnlyFlow

+ Source video

Output

Figure 1. **OnlyFlow** controls the generation of video with text and motion of a video input, synthetically generated or not. We strongly encourage readers to check our supplemental content for video results that are not well represented by still images.

## Abstract

*We consider the problem of text-to-video generation tasks with precise control for various applications such as camera movement control and video-to-video editing. Most methods tacking this problem rely on providing user-defined controls, such as binary masks or camera movement embeddings. In our approach we propose OnlyFlow, an approach leveraging the optical flow firstly extracted from an input video to condition the motion of generated videos. Using a text prompt and an input video, OnlyFlow allows the user to generate videos that respect the motion of the input video as well as the text prompt. This is implemented through an optical flow estimation model applied on the input video, which is then fed to a trainable optical flow encoder. The output feature maps are then injected into the text-to-video backbone model. We perform quantitative, qualitative and user preference studies to show that OnlyFlow positively compares to state-of-the-art methods on a wide range of tasks, even though OnlyFlow was not specifically trained for such tasks. OnlyFlow thus constitutes a versatile, lightweight yet efficient method for controlling motion in text-to-video generation.*

## 1. Introduction

Progress in generative AI has made tremendous progress thanks to the rise of diffusion models [20] and colossal datasets [33]. Previously, research on generative models focused mainly on generating images without any direct control, but today's generative models have shifted towards integrating text to control the generation process and to facilitate interaction with the user. This has started to revolutionize the creation industry (*e.g.*, professional entertainment, advertising, art) by offering increased productivity and creativity. However, today's synthesis pipelines follow the text-based generative paradigm, which does not provide precise user control because the text modality often limits user expression.

In text-to-video (T2V) synthesis, the controllability problem is more pronounced than text-to-image (T2I), since the generation process has an additional axis: the temporal dimension. Yet, the generative community has made some progress in adding more control to T2V pipelines. Recent work has focused on providing more precise control by using external conditioning data for T2V models [10, 13, 23, 40, 49]. For example, users can edit parts of the generated content using masks, apply styles from existing images, or even switch between content, layout, and style conditioning. Despite that, the underlying challenge in

video creation is related to the complexity of motion synthesis. By adding motion constraints, one can reduce the ambiguity inherent in video synthesis. This allows for better motion modeling and improved ability to manipulate generated content for personalized creations.

In this paper, we introduce OnlyFlow, a simple solution for controlling video generation by directly integrating motion cues. In a nutshell, OnlyFlow constrains the generated video to mimic the input motion. More specifically, our proposed approach exploits optical flow information and uses it to guide the generation process using a small externally trained optical flow encoder. However, unlike most methods that use optical flow, our method works together as a T2V or video-to-video (V2V) model, as seen in Figure 1. On the one hand, OnlyFlow can use synthetically generated flows to simulate any movement, thus fixing the content to the text while respecting the input motion. On the other hand, we can use the flow of an input video and translate it to the generated one.

To summarize, our contributions are threefold. (1) We propose OnlyFlow, a simple and novel motion-guided strategy for video synthesis that enables motion conditioning from given reference videos based on optical flow representations, allowing motion transfer across videos. (2) We conduct extensive empirical studies to validate OnlyFlow from both quantitative and qualitative perspectives. (3) We show the versatility of our approach in various video generation tasks, where several experiments demonstrate its use case.

## 2. Related work

### 2.1. Text-to-video diffusion models

Recently, T2V synthesis has been used to create realistic videos for creative applications. These applications build on generative backbones such as AnimateDiff [15], CogVideoX [22], or VideoCrafter [8]. A common technique for training video synthesis models is to use powerful pretrained image generators [32]. To do this, most approaches have introduced temporal layers [3, 4, 8, 15, 19, 35] in the form of temporal attention layers and 3D convolutions, and fine-tuned these models on large video datasets [1, 41]. Other approaches [5, 24, 44] operate directly on the features of the T2I model. Finally, some methods train their method from scratch [6, 14, 17, 21, 22, 46]. These works typically tune their model in two steps: using a large amount of low quality data (videos and images) and then fine-tuning the model with high quality but scarce videos. However, all these models have a common flaw: motion controllability is limited. In this paper, we improve this aspect and build OnlyFlow on top of the popular AnimateDiff [15] model, but it can be easily extended to any T2V.

### 2.2. Controllable video generation

Recent developments in controllable image generation have sparked interest in bringing similar control mechanisms to video generation. Researchers have introduced various control signals to guide the creation of videos, such as specifying the initial frame [16, 40], defining motion trajectories [34, 40, 49], focusing on particular motion regions, controlling specific moving objects [42, 43], or introducing images as regularizers to enhance video quality and improve temporal relationship modeling [25, 40]. Additionally, many works propose motion-specific fine-tuning approaches [13, 15, 44]. While showing promise in capturing nuanced motion, they involve demanding training processes [15, 44] or costly inversion processes [13, 23, 36], risking degrading the model's performance, while not been user-friendly. Finally, controlling camera motion during video generation is another area of focus. [18, 42] introduce camera representation embeddings to manipulate camera poses. But, these may be limited by the number of camera parameters they can handle. While camera control seems promising in theory, it is limited in terms of actual creation, as creating any complex motion is difficult.

Overall, while significant progress has been made in controllable video generation, providing precise control over motion of intrinsic object and camera dynamics without compromising model performance remains an ongoing challenge. We believe that OnlyFlow opens a direction towards achieving this goal.

### 2.3. Optical Flow as Motion Priors for Video Synthesis

Closest to our work are the studies that integrate optical flow into the generation pipelines. Undoubtedly, optical flow is an intuitive signal to represent motion in videos. Therefore, many works have explored the introduction of optical flow to guide the generation towards a specific motion, especially in V2V and I2V generation [9, 26, 27, 45]. Common techniques include warping the first frame of the video and postprocessing it with the generative model. In addition, many methods include depth estimation maps to further constrain the generation. All these methods have in common that they use the optical flow mainly as a guiding tool for the V2V or I2V task using sophisticated and complicated mechanisms. In contrast, no other approach allows for generating both unrelated patterns from motion (*e.g.*, waves) and content from prompt (*e.g.*, field of flowers), while being simple.

## 3. OnlyFlow framework

### 3.1. Diffusion Model for Video Generation

Text-to-video generation involves the creation of coherent and high-quality videos conditioned on textual prompts. Due to the high dimensionality and temporal complexity

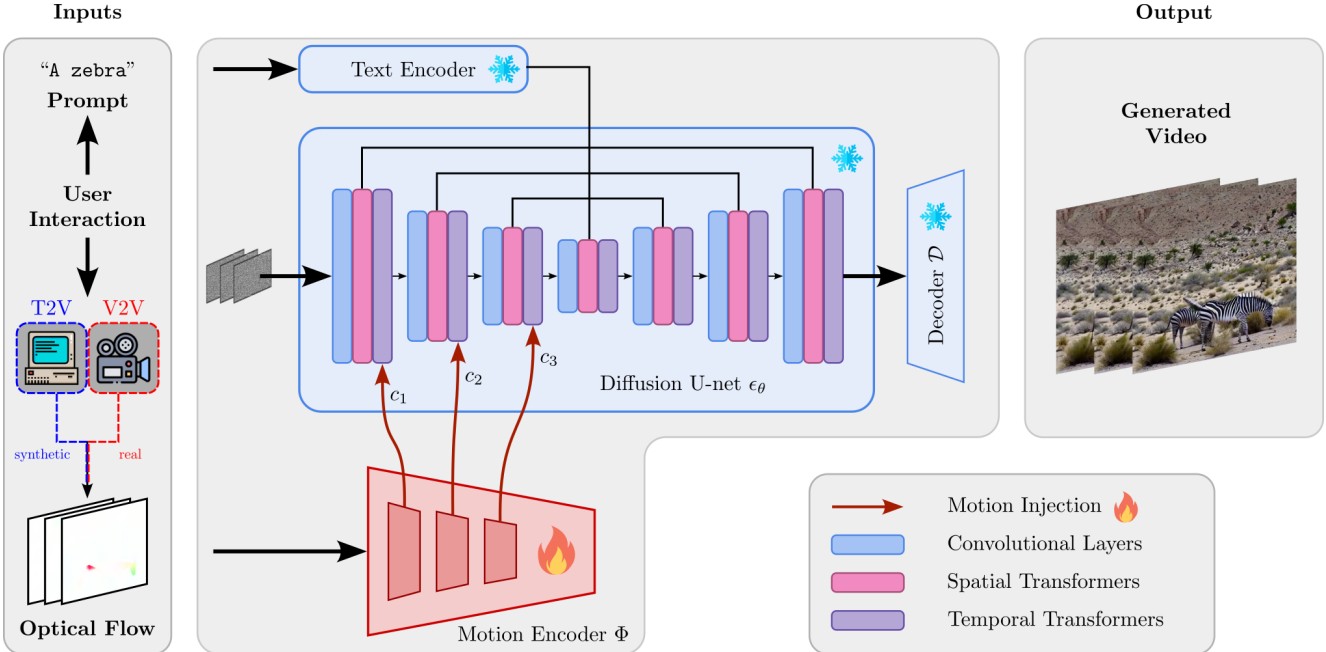

Figure 2. Overview of OnlyFlow. Inputs are i) a tokenized and encoded text prompt, ii) noisy latents for the diffusion model and iii) the optical flow of an input video. The latter is fed through a trainable optical flow encoder which outputs features maps that are injected in the diffusion U-net. We experiment with several injection strategies, for illustration purposes we only show the injection in temporal attention layers of the U-net. The U-net is kept frozen during training. The output generated video matches the input prompt and motion.

of video data, recent approaches [32] compress the spatial resolution of the input videos $V$ into a lower dimensional latent representation $z$ using an encoder $\mathcal{E}$, *i.e.*, $\mathcal{E}(V) = z$. To project the latent codes back into pixel space, a decoder $\mathcal{D}$ operates on this latent representation to reconstruct the videos.

In this latent space, denoising diffusion probabilistic models (DDPMs) [20] are employed to approximate the latent distribution of video data. The forward diffusion process gradually adds Gaussian noise to the latent variables over $T$ timesteps, producing noisy latents $\mathbf{z}_t$ using the formulation:

$$\mathbf{z}_t = \sqrt{\bar{\alpha}_t}\mathbf{z}_0 + \sqrt{1 - \bar{\alpha}_t}\epsilon, \quad \epsilon \sim \mathcal{N}(0, \mathbf{I}) \qquad (1)$$

where $\bar{\alpha}_t = \prod_{i=1}^{t} \alpha_i$, and $\alpha_t$ controls the noise schedule. The reverse diffusion process aims to recover $\mathbf{z}_0$ by learning a neural network $\epsilon_\theta$ that predicts the added noise at each timestep $t$.

In this work, we build our architecture on top of AnimateDiff [15]. To effectively capture spatial and temporal dynamics, AnimateDiff first expands Stable Diffusion to video synthesis, inflating the 2D U-net architecture into a 3D U-net. Secondly, it adds temporal convolutions and attention mechanisms to capture temporal dynamics. As a result, the diffusion model factorizes its internal operations over space and time, meaning that each layer operates only on the space or time dimension, not both at the same time.

Formally, setting as $B$, $F$, $H \times W$, and $C$ as the batch size, number of frames, spatial dimensions, and channel dimension, respectively, each feature map represents a 5D tensor of $B \times F \times H \times W \times C$ that can be fed into the following layers:

- **spatial layers**: each old 2D convolution layer as in the 2D U-net is extended to be space-only 3D convolution over $H \times W \times C$. Each spatial attention block remains as attention operating on individual frames.
- **temporal layers**: a temporal attention block is added after each spatial attention block. It performs attention over $F$. The temporal attention block is important to capture good temporal coherence.

### 3.2. Optical flow conditioning in T2V Architecture

In this paper, we propose OnlyFlow to modify this traditional T2V scheme by adding motion constraint. Fig. 2 illustrates an overview of our model architecture. More precisely, for enhanced controllability in video generation, we propose to condition the video generation on an optical flow signal $f$. Here, $f$ can be synthetically created or generated through standard optical flow extraction techniques [37].

The architecture of OnlyFlow extends AnimateDiff by incorporating a motion encoder that processes optical flow input and extracts motion features at various scales. We then proceed to inject these signals into the temporal attention layers of the encoder of the U-net. Let us describe for-

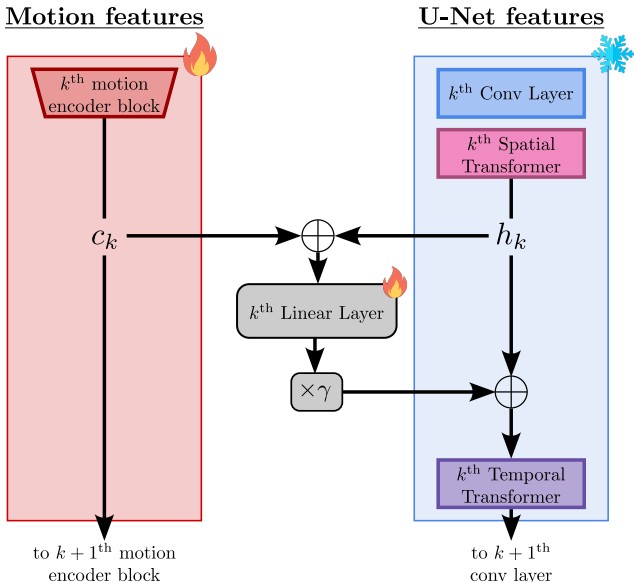

Figure 3. Injection strategy of the encoded optical flow conditioning $c_k$ from the optical flow encoder into the temporal attention layers of the $k$-th block of the U-net.

mally both the motion encoder architecture and the feature injection process:

- **Motion feature architecture**: inspired in T2I-adapters [28] and CameraCtrl [18], our motion encoder architecture $\Phi$ first unshuffles [7] the input flow $f$ before applying a convolutional layer. It then processes the features into a sequence of spatial resnet blocks and temporal attention operations. Finally, we store the motion features $\Phi(f) = \{c_k\}_{k=1}^{4}$ after each temporal operation to inject them into the U-net.

- **Motion feature injection**: As seen in Fig. 3, we combine the latent features $h_k$ of the $k^{\text{th}}$ block of the U-net with the control signal from motion features $c_k$ through element-wise addition. Then, the combined features are processed by a linear layer and scaled using a parameter $\gamma$. Finally, the output is summed back to $h_k$ and it is directly feed back into the temporal layers of the U-net architecture. Formally, the complete injection operation for the layer $k$ is:

$$h'_k = h_k + \gamma \, \text{Linear}_k \left( c_k + h_k \right). \quad (2)$$

Here, the parameter $\gamma$ effectively controls the influence of the optical flow from the input auxiliary video onto the generated video. It is the main parameter to boost or decrease the auxiliary motion injection over the generated output. We call $\gamma$ the **optical flow conditioning strength**.

### 3.3. Training OnlyFlow

To train our proposed approach, we follow the original formulation of diffusion model training, but include our control signal. More specifically, we optimize the parameters $\phi$ of the motion encoder $\Phi$ by minimizing the standard diffusion model loss

$$\mathcal{L}(\phi) = \mathbb{E}_{(V,p),t,\epsilon} \| \epsilon - \epsilon_\theta(z_t, t, p, \Phi(f)) \|, \quad (3)$$

where, $z_0 = \mathcal{E}(V)$, $z_t$ is created using Eq. 1, $p$ is the corresponding textual description of $V$, and $f$ is the output of an optical flow predictor $\mathcal{T}$ using as input $V$, *i.e.*, $f = \mathcal{T}(V)$. Of course, during the training phase, we freeze the parameters of AnimateDiff. Only the motion encoder and the merging linear layers in the attention blocks are trained. Additionally, we set the optical flow conditioning strength to $\gamma = 1.0$.

## 4. Experimental Results

### 4.1. Implementation details

**Dataset and Preprocessing** To train OnlyFlow, we used the WebVid dataset [1], which consists of 10.7M video-caption pairs, totaling 52K hours of video content. The majority of these videos have a length between a few seconds and 1 min, with 100k to 500k pixels by frame. All videos are spatially resized to $256 \times 384$ pixels, and $F = 16$ consecutive frames are randomly chosen, matching the training practice used in AnimatedDiff [15].

**Learning setup** We train OnlyFlow for approximately 20 hours using 8 A100 GPUs. The training process involved a batch size of 32, a constant learning rate of $1 \times 10^{-4}$ with the Adam optimizer. The model consists of 198M parameters in the optical flow encoder $\Phi$ and 19M for the injection layers in the U-net attention. To extract the optical flow $f$ during the training phase, we employed the commonly used RAFT-Large [37] model due to its fast inference and widely usage in computer vision tasks.

**Experimentation Goals** As a quick recap, our main goal is to transfer the motion cues from a video to the generated one while remaining realistic and faithful to the textual prompt. Therefore, we assess our model quantitatively using three main components: realism, flow fidelity, and textual alignment.

First, to ensure that our model produces *realistic* videos, we report the mean Fréchet Video Distance (FVD) [38] across frames. This metric evaluates the temporal coherence of generated videos compared to real ones from a distributional perspective. We follow standard practices proposed by Unterthiner *et al.* [38] to compute the FVD.

Next, we assess the *flow fidelity* between the generated video and the prompted one, *i.e.* whether the produced video contains the same motion patterns as the input. To evaluate this, we measure the absolute pixel-wise difference between the optical flow of the input video and the generated data.

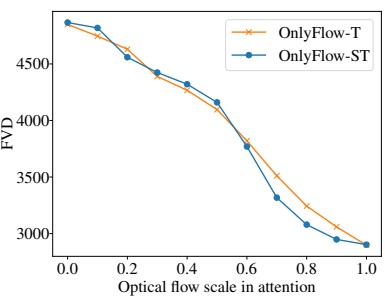 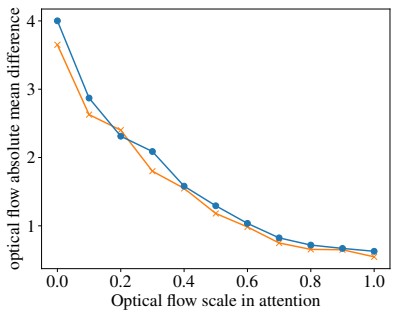 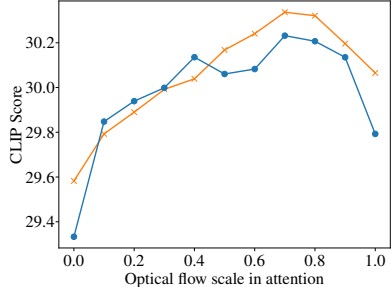

(a) Fréchet Video Distance (lower is better) between source generated video.

(b) Mean absolute difference in optical flow between source and generated video.

(c) CLIP score similarity between prompts and generated video.

Figure 4. Metrics computed on OnlyFlow-generated videos for both feature maps injection strategies (OnlyFlowT (orange) and OnlyFlow-ST (blue).) for different values of $\gamma$. The motion realism depicted by fig.(a) improves with $\gamma$, as well as the optical flow similarity in fig.(b). Fig.(c) shows us that the CLIP score does not deteriorate and therefore that the video respect the prompt just as well

Finally, to assess the *textual alignment* between the input text and the generated video, we adopt the CLIP similarity (CS). Specifically, we focus on the expected CS across frames. To study the CS, we used a OpenAI Vision Transformer (ViT) architecture as the image encoder, with a patch size of 16x16 pixels [30].

### 4.2. Evaluating OnlyFlow

**Quantitative assessments** To empirically validate OnlyFlow, we perform extensive testing using a subset of 70,000 unseen videos from the WebVid [1] dataset to generate the optical flow to condition our proposed model. All the assessment was performed on generated videos with $F = 16$ frames at a resolution of $512 \times 512$ pixels.

As for the evaluation metrics, we validate our model using the FVD, flow fidelity, and textual alignment metrics, previously detailed. Finally,to analyze the impact of our proposed motion module, we evaluate several optical flow conditioning strengths $\gamma$ (Eq. (2)), as well as using two OnlyFlow variants: OnlyFlow-T, referring to injecting the feature in the temporal layers (*i.e.* our OnlyFlow base model), and OnlyFlow-ST, where we inject the motion features in both spatial and temporal layers.

First, we discuss the FVD metric in Fig. 4a. Interestingly, we notice that as we increase optical flow conditioning strength $\gamma$, the FVD decreases. We posit that this is explained by the addition of real data as a condition to video generation. Real data brings realistic motion which will end up in being used by OnlyFlow, which in turn will be interpreted as accurate by the FVD metric. In this generative process, enforcing the optical flow of an input video with realistic motion naturally improves the FVD.

Next, we review the motion fidelity in Fig. 4b, measured as the absolute difference in optical flow. As expected, the difference between optical flow of the input video and optical flow of the generated video decreases as the optical

| Ref. Method | Judgements in our favor (%) |
|---|---|
| TokenFlow [13] | 54.5% |
| Control-A-Video | 71.1% |
| RAVE [23] | 62.1% |
| Gen-1 [12] | 60.6% |
| VideoComposer [40] | 63.6% |

Table 1. **User study: preference (%) between OnlyFlow and reference methods.** We report the percentage of judgments in favor of OnlyFlow w.r.t. each baseline.

flow conditioning strength increases. This finding implies that the optical flow of the generated video resembles the one of the source video, effectively showing that OnlyFlow achieves our main objective of transferring the motion cues.

Next, we examine the textual alignment results in Fig. 4c. The outcome of this experiment show that the model produces stable CLIP scores when using any level of optical flow conditioning. This suggests that the content generated by the model is aligned with the text to the same extent, whether the motion cues are used or not.

Finally, we opt to insert the optical flow conditioning representations into the temporal attentional blocks. This decision is motivated by the ability of the temporal attention layer to capture temporal dependencies, which is consistent with the sequential and causal nature of optical flow. From previous empirical results, we find that injecting the motion feature into the spatial and temporal layers together has no advantage over introducing it into the temporal layers, justifying our choice of design injection. For the remainder of the paper, all results were performed using OnlyFlow-T.

**User preference study** While automatic metrics approximate the behavior of T2V pipelines, the gold standard for evaluating generative models today is a human study. To this end, we use the Two-alternative Forced Choice (2AFC)

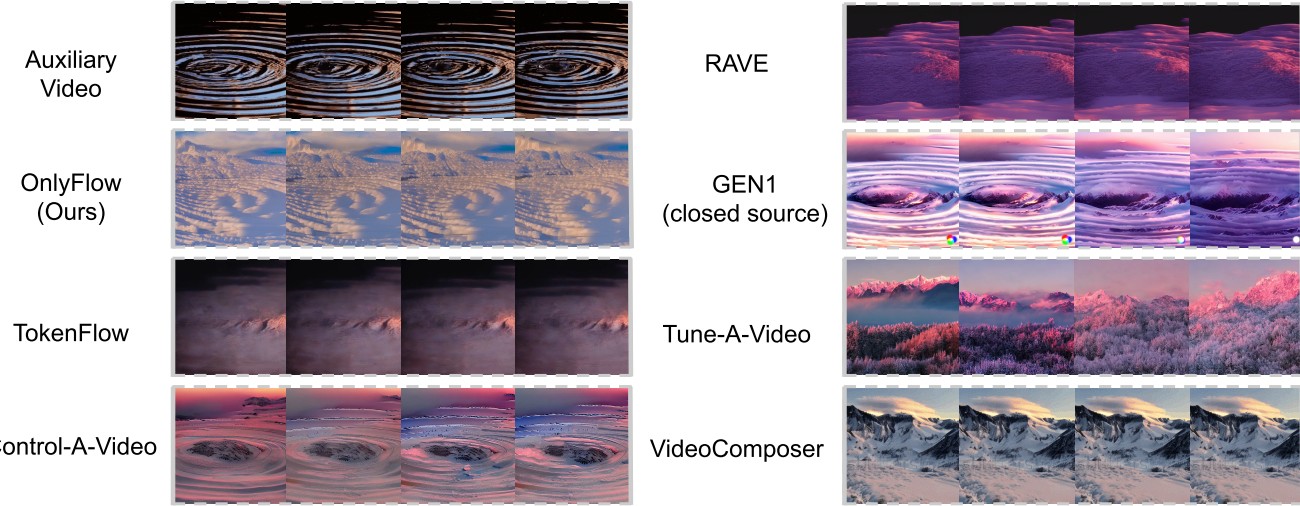

Prompt: "A snow-covered mountain in the dawn, with pink clouds hovering low"

Figure 5. Qualitative comparison of video-to-video generation models. Videos are generated using same text prompt and input video. OnlyFlow exhibits a superior combination of motion fidelity and image realism. It positively compares to approaches that uses depth map (RAVE, VideoComposer, Control-A-Video), is comparable to Gen-1 [12]'s temporal coherence and VideoComposer [40]'s image quality.

protocol for text-driven video editing [2, 11, 13, 29, 47]. Participants are presented with the input video, our results, and a baseline, and are asked to decide which video better matches the text prompt, which video better preserves the motion of the input video, and which video they prefer overall. We collected 300 user judgments. As seen in Tab.1, our method is consistently preferred over all baselines.

**Qualitative comparison with state-of-the-art video-to-video generation models** For the loosely-defined problem of video-to-video editing and customization, we show that our OnlyFlow model performs better for tasks that require only the motion of the input video. The visual comparison in Fig. 5 reveals that OnlyFlow can generate animations whose movements are close to the input video, given a different textual prompt. In this comparison, the rippling waves effect of the auxiliary video struggle to be accurately represented in other models, except for Control-A-Video and Gen-1. We notice that the prompt adherence is also better followed in our approach, Gen-1, VideoComposer and Tune-A-Video. Nevertheless, the latter two models did not achieve to take motion inspiration from our auxiliary video and Gen-1 seems to stack mountains and ripples visuals. Compared to other work, such as Tune-A-Video [44], our approach is lightweight. On top of that, it does not require any costly DDIM [36] inversion process used in TokenFlow [13] and RAVE [23].

**Comparison with modern camera-movement controlled video generation** In recent work, such as MotionCtrl [42]

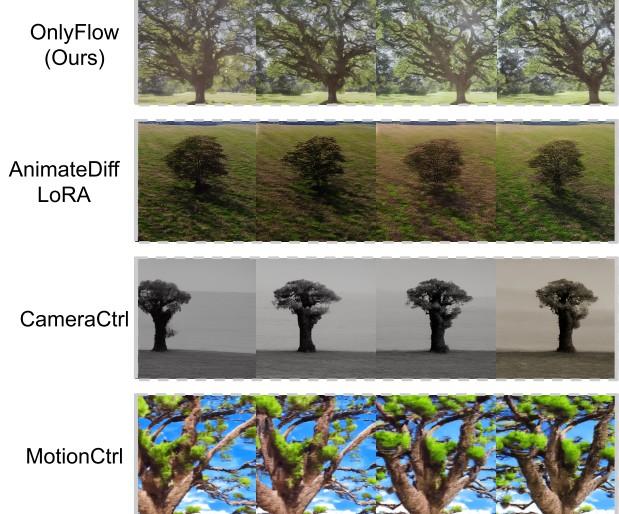

Prompt: "a majestuous tree, a grassy plain"
Task: Camera movement > Pan left

Figure 6. Qualitative comparison on camera-movement controlled video generation. All videos are generated using the same text prompt, camera trajectory (pan-left), and seed when available. Without having trained for this task, our OnlyFlow model achieve the same camera control capability as other camera-movement approaches trained on this task.

and CameraCtrl [18], the conditioning of camera motion is achieved with a camera position encoding, while the AnimateDiff [15] motion conditioning uses a LoRA trained on

Prompt: "A drop of water jumping in purple ink"

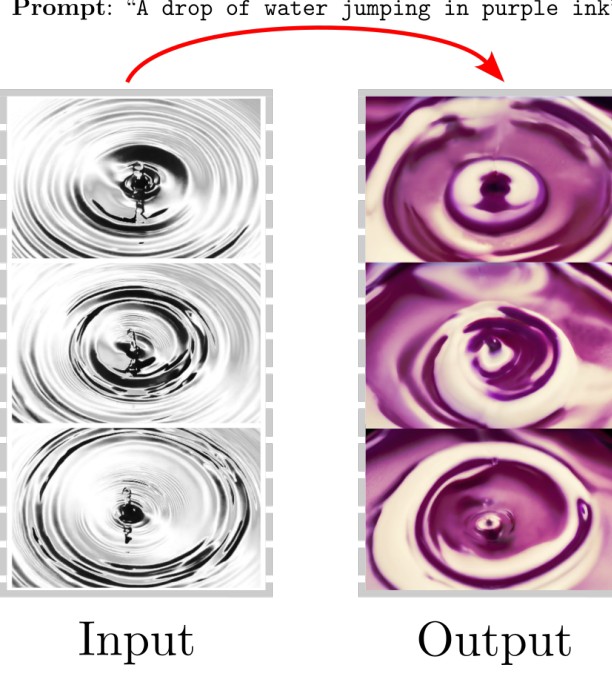

Input                    Output

Figure 7. One of our interests is the creation of artistic videos. Here we show an example of our results, which we projected in an internationally renowned art venue (unnamed for anonymity).

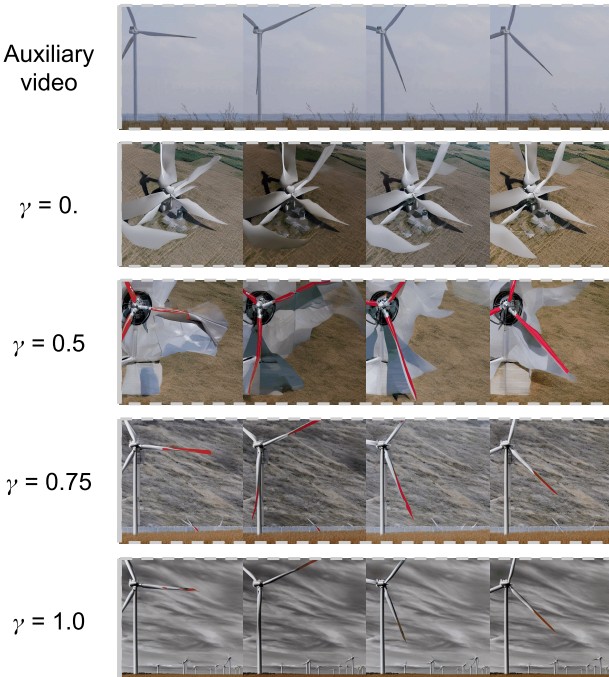

Prompt: "wind turbine in a field, cloudy weather"

Figure 8. Illustration of OnlyFlow's optical flow conditioning strength impact on generated videos. Setting the $\gamma$ scale at which the features from the input video are injected sets the influence of its motion on the generated video. The videos presented in each row are obtained for increasing values of $\gamma$ between 0 and 1.0. We observe a progressive alignment of the generated video motion and wind turbine position on the auxiliary video

the temporal layer for a given motion. For camera displacements, OnlyFlow provides a flexible alternative for modeling camera movement. To do this, we create an artificial optical flow to describe the desired motion or provide a video with the desired motion. In Fig. 6 we show a specific scenario where we prompted our model with a constant horizontal movement to the left. The generated results clearly show that the generation is on par with the controllability of specific approaches such as CameraCtrl or MotionCtrl. Note that we did not observe satisfactory results with AnimateDiff's motion LoRA, both in terms of aesthetics and expected camera movements. This shows the versatility of our model.

### 4.3. OnlyFlow for Art

As previously show, OnlyFlow has many appealing properties. Consequently, our model may be applied to many creative scenarios. In particular, we are interested in using OnlyFlow for artistic applications. Using the natural motion present in video footage allows the generation of videos that would require tremendous editing efforts to be created without OnlyFlow. In Fig. 7 and the supplementary video, we provide an artistic video we created during the project, that incorporate upscaling with ESRGAN [39] and frame interpolation with FILM [31], which by themselves showcase the type of outputs that can be done using OnlyFlow. Moreover, OnlyFlow is readily compatible with extensions

of AnimateDiff and Stable-Diffusion that are often used in artistic creation context such as IP-Adapter [48], which can further be used to improve the quality and controllability of the generated videos. Finally, we did an art exposition based on OnlyFlow on internationally recognized art-venues. Yet, to remain in anonymity, we will not go further in details. Instead, for the camera-ready, we will provide an in-depth description of our exposition.

### 4.4. Analyzing OnlyFlow

**Flow strength control** Our model can be utilized with various values of conditioning strengths $\gamma$ to control of the movement in the generated video. In Fig. 8, we present an illustration of this phenomenon using the fixed textual prompt `Wind turbine in a field, cloudy weather`. In this particular example, no rotative movement is observed when OnlyFlow is disabled ($\gamma = 0$). As the conditioning strength increases, the rotative axes align, followed by the blade shape and length, with small wind turbines appearing in the background on tall grass in the auxiliary video. Manually tuning $\gamma$ during the inference can

help the user to better express the desired motion.

**Semantic alignment**   We noticed an interesting capability of semantic alignment between the descriptive information contained in the optical flow fed into the model and the given prompt. As shown in Fig. 9, the optical flow inadvertently captures object shapes. When there is a match between the input video motion concept and the prompt, the model successfully maps the zones of interest to be generated. This ability is similar to other conditioning models such as ControlNet [50] and T2I-Adapter [28], but our method does not require manual control map creation.

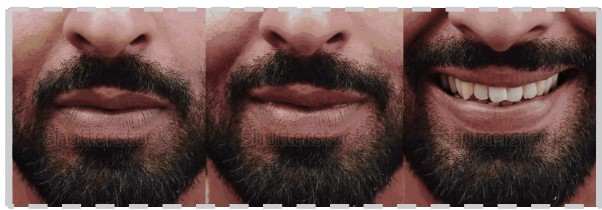

(a) Frames from the auxiliary video of which optical flow is extracted

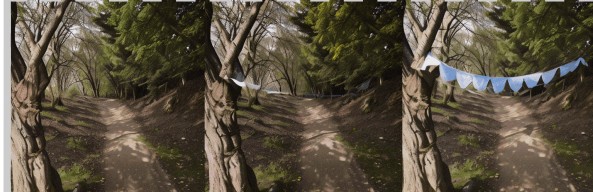

(b) Frames from corresponding generated video

Figure 9. Example of semantic alignment between optical flow and prompt. With the prompt "`Trees in forest`", our OnlyFlow model generates a clothesline corresponding to the teeth of the smile.

## 5. Conclusions

In this paper, we introduced our OnlyFlow method, which improves on T2V models such as AnimateDiff by conditioning it on the motion extracted from an input auxiliary video. The optical flow of the auxiliary video is then passed to a trainable flow encoder connected to the T2V model. The flow encoder outputs features maps which are injected into the attention layers of the T2V's U-net. The generated video is encouraged to accurately follow the motion of the input video. While other methods allow conditioning by movement (with motion masks or user-defined vectorial camera movements), OnlyFlow allows generating any video with a text prompt and the motion of an auxiliary video. In our experiments, we demonstrate the efficiency of OnlyFlow for various applications such as camera motion control (panning, zooming, etc.) and video editing/style transfer. Without necessarily being trained for such applications, OnlyFlow compares positively to other state-of-the-art methods that have been specifically trained for such

applications. We also carried out a user study which confirmed the interest of our method in generating very interesting videos, demonstrating that their movement convincingly and coherently follows the input reference.

**Limitations**   One clear limitation of our work is photorealism obtained in the generated videos. While T2V models like AnimateDiff [15] are powerful generative models, their generation remains poor for most complex prompts depicting scenes with various objects and subjects or scenes with movements that might not appear in the training set. The resolution of the generated video is also limited by the underlying model.

Another limitation is that while optical flow as motion conditioning is useful and efficient in most cases as shown in our experiments, optical flow is not the only way to estimate motion in a video. We could use other motion conditions. For instance, optical flow does not always allow separate camera movements from object movements. Thus, in order to better model the motion of an input video, other methods could be investigated, such as Kalman Filtering, Block or Feature Matching, Homography-based Motion estimation. A combination of such motion estimations could lead to better results, and are left for future work.

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
