# OpenReview forum: "OnlyFlow: Optical Flow based Motion Conditioning for Video Diffusion Models"
_thecvf.com/CVPR/2025/Workshop/CVEU — CVPR 2025_

### Official Review · Reviewer_mq8q · 2025-03-15
**OnlyFlow: Optical Flow based Motion Conditioning for Video Diffusion Models**

**Rating:** 3
**Confidence:** 4

**Review:**

Pros:
1. The paper utilizes optical flow information to generate videos that are motion-consistent with the input video and conform to textual cues.
2. The paper provides a comprehensive assessment of OnlyFlow through quantitative, qualitative and user preference studies.
3.OnlyFlow performs well in a variety of video generation tasks, such as camera motion control and video-to-video editing. This demonstrates OnlyFlow's broad applicability and flexibility.

Cons:
1. Optical flow estimation does not always accurately distinguish between camera motion and object motion. The paper uses only this one constraint, which may lead to less accurate motion control of the generated video in some cases.
2. Extracting the optical flow and training the optical flow encoder requires additional computational resources. This may increase the training and inference cost of the model.
3. Optical flow is a common prior in video algorithms, and many existing video processing tasks (e.g., video interpolation, motion segmentation, etc.) use similar designs. OnlyFlow's design and use of the optical flow prior does not introduce significant novelty.

---

### Official Review · Reviewer_tNz4 · 2025-03-17
**Concise model framework but the strong dependence on optical flow limits the model's capabilities.**

**Rating:** 2
**Confidence:** 5

**Review:**

The paper introduced a motion (represented by optical flow) guided video editing method named OnlyFlow. By using optical flow from a generated video, through T2V or V2V, authors propose a motion feature encoder to inject the motion features into U-net blocks of a video diffusion model through linear layers. Comparing with a video generation method, OnlyFlow is more like a video editing method since it relies on optical flow which is extracted from an existing video (generated or ready-made). In addition to optical flow, OnlyFlow also accepts text prompt as an additional condition to control the resulting video.

### Strengths:

* Concise model structure: The injection of optical flow features, into Diffusion U-Net blocks, is quite straightforward.
* Good reproductivity: The training data and base model is WebVid and AnimateDiff respectively, both of which are open-sourced. The trainable parameters are affordable and the training cost is not very high.
* Good transferability: The trainable parameters are mainly from additional motion encoder and the corresponding linear layer for motion injection. Thus, OnlyFlow can be easily tranfered to another base video diffusion model.

### Weaknesses:

* Considering the inputs of OnlyFlow (*text prompt* and *optical flow* from an existing video), this is more like a video editing method than a video generation method. In the paradigm of video editing, optical flow is a quite strict dependence since it limits the whole motion of every objects or humans. My concern is from the controlling ability of text prompt. Once an optical flow is given, from an existing video, can we do these things like other video editing methods: add/delete objects; change shape/color of an object; change background?
* In stead of optical flow, this work can be better if we can use any other condition, like deption or pose sequences, to work with the main framework of OnlyFlow.
* Optical flow, which is necessarily required by OnlyFlow, is not an *intuitive signal* (Line #129) because it can barely be produced without an existing video. On the contrary, *use-defined control* (Line #004), like dragging gestures, is more intuitive.
* The resulting videos are not very good. In Figure 1, the left part shows the resulting video frames produced by OnlyFlow. Neither objects nor humans appear, we can hardly evaluate its quality.
* The comparisons with other methods are not fair. OnlyFlow relies on *an artifical optical flow* (Line #369) while other methods do not have this limitation.

---

### Official Review · Reviewer_RHD9 · 2025-03-23
**Brief review**

**Rating:** 4
**Confidence:** 4

**Review:**

Advantages
1. The proposed method introduces a straightforward yet effective way to condition text-to-video generation using optical flow as a motion prior.
2. OnlyFlow is compatible with existing T2V frameworks like AnimateDiff and can be used for both text-to-video and video-to-video tasks without additional training.

Improvement
1. The authors themselves acknowledge that optical flow has limitations (e.g., in separating camera and object motion). Exploring or integrating complementary motion cues (e.g., scene flow, pose trajectories) could further improve performance.
2. Although this paper explores a training-free algorithm, its performance still falls significantly short compared to current state-of-the-art video foundation models such as CogVideoX and HunyuanVideo. Improving the visual quality and the range of motion in the generated videos remains an important direction for future work.
3. This paper explores the field of video generation, yet surprisingly lacks video-level visualizations. Including corresponding video examples, comparisons, and failure case analyses in the supplementary materials would greatly enhance the credibility and persuasiveness of the work.

---

### Decision · Program_Chairs · 2025-03-25

**Decision:**

Accept

**Comment:**

The paper introduces OnlyFlow, an optical-flow-based approach to condition motion in video diffusion models. Reviewers highlighted its practical value, simplicity, and good compatibility with existing methods. However, they noted limitations such as strict dependence on optical flow, limited novelty, and insufficient exploration of alternative motion cues.

Overall, the paper is accepted. Authors should clearly address reviewer feedback by discussing optical flow limitations and including video-level visualizations in the camera-ready version.